# Lateral orbitofrontal neurons acquire responses to upshifted, downshifted, or blocked cues during unblocking

**Nina Lopatina[1,2], Michael A McDannald[3], Clay V Styer[1], Brian F Sadacca[1], Joseph F Cheer[4], Geoffrey Schoenbaum[1,5,4]***

[1]Intramural Research Program, National Institute on Drug Abuse, Baltimore, United States; [2]Program in Neuroscience, University of Maryland School of Medicine, Baltimore, United States; [3]Department of Psychology, Boston College, Boston, United States; [4]Department Anatomy and Neurobiology, University of Maryland School of Medicine, Baltimore, United States; [5]Department of Neuroscience, Johns Hopkins University, Baltimore, United States

**Abstract** The lateral orbitofrontal cortex (lOFC) has been described as signaling either outcome expectancies or value. Previously, we used unblocking to show that lOFC neurons respond to a predictive cue signaling a 'valueless' change in outcome features (McDannald, 2014). However, many lOFC neurons also fired to a cue that simply signaled more reward. Here, we recorded lOFC neurons in a variant of this task in which rats learned about cues that signaled either more (upshift), less (downshift) or the same (blocked) amount of reward. We found that neurons acquired responses specifically to one of the three cues and did not fire to the other two. These results show that, at least early in learning, lOFC neurons fire to valued cues in a way that is more consistent with signaling of the predicted outcome's features than with signaling of a general, abstract or cached value that is independent of the outcome.

***For correspondence:** geoffrey. schoenbaum@nih.gov

**Competing interests:** The authors declare that no competing interests exist.

## Introduction

The orbitofrontal cortex (OFC) is often described as signaling either an outcome expectancy, implying a knowledge of the features of the impending outcome (*Schoenbaum et al., 1998*; *Delamater, 2007*; *Luk and Wallis, 2013*; *Ostlund and Balleine, 2007*; *Steiner and Redish, 2012*), or a value that exists independent of those features (*Levy and Glimcher, 2012*; *Padoa-Schioppa, 2011*). Notably, while support for abstract value encoding comes largely from studies employing economic decision-making procedures (*Plassmann et al., 2007*; *Padoa-Schioppa and Assad, 2006*; *Levy and Glimcher, 2011*), it has been suggested that the 'economic' value revealed by these procedures is the same as that underlying well-controlled OFC-dependent behavioral effects such as reinforcer devaluation or sensory preconditioning (*Padoa-Schioppa, 2011*). Notably, in these settings, the role of the OFC (*West et al., 2011*; *Jones et al., 2012*; *Pickens et al., 2003*) cannot be strictly explained by signaling of an abstract or outcome-independent value. Indeed, the behavior that requires OFC reflects a value that is inextricably linked to, or an attribute of, the predicted outcome - the reward in the case of reinforcer devaluation or the primary conditioned stimulus in the case of sensory preconditioning. This implies that OFC allows access to a representation of the outcome through which the value can be inferred or calculated on the fly, at the time the behavior is engaged. Thus, while the aforementioned correlates could reflect the final common calculation, it is equally reasonable that they might reflect the specific associative representations that are activated as part of this calculation. Note this is even true of coding changes with subjective preference (*Padoa-Schioppa and*

*Assad, 2006*, supplemental), since the perception of attributes is influenced by valuation or preference (e.g., a nice warm cup of hot chocolate while skiing in the winter versus the same cup at the beach in the summer).

We recently tried to distinguish between these possibilities by using Pavlovian blocking to strip away or 'block' the value portion of the outcome during learning, while leaving unblocked the outcome's unique sensory and other features (*McDannald et al., 2014*). We did this by pairing a 'target' cue with a rewarding outcome in the presence of a cue that had been previously trained to predict a differently-flavored, but similarly-valued outcome. When this is done, the previously conditioned cue predicts the value that is common to the two outcomes, but does not predict the unique features that distinguish the new outcome (note features are not limited to sensory properties, but might include the outcome timing, location, temperature, size, number, etc.). The target cue acquires associations with the unique features of the new outcome but not with its general value (*Burke et al., 2008*; *Rescorla, 1999*). Notably such unblocking is dependent on the OFC, whereas value-based unblocking is not (*McDannald et al., 2011*). In this prior study, we found that OFC neurons responded strongly to this valueless target cue (*McDannald et al., 2014*). In fact, they responded as strongly to this cue as they did to a control cue that was paired with an additional reward.

While this result was perhaps not surprising, given prior evidence that OFC neurons signal both value as well as sensory features of predicted rewards (*Padoa-Schioppa and Assad, 2006*), we were intrigued by the similarity in the strength of these representations. After all, if OFC is primarily concerned with representing value or at least with encoding things with biological significance, then why was the neural representation of a cue signaling a valueless change in reward features so similar to the representation of a cue that actually predicted a more valuable reward? Furthermore, we did not find any correlation in the firing of the value-predicting neurons across other cues present in the task that also predicted reward and value. This suggested to us that the neural activity to this cue was likely not only representing value but instead was representing the unique features that led to the value prediction in the case of that cue; e.g., the additional reward.

Here we have designed a follow-up experiment to test this question. In this experiment, we have copied the design of the prior study (*McDannald et al., 2014*), recording OFC neurons during unblocking with two changes. The first change was to replace our reward identity shift with a downward shift in value to directly test whether responses to additional reward in the prior study were related to value, which should be reversed in this condition, or some other aspect of the additional reward, which might be unrelated. The second change was to deliver a single drop of reward rather than multiple drops to minimize differences in features between our reward manipulations. We also shifted the size of the outcome rather than the number because number downshift unblocking results in excitatory behavior while a concentration shift does not (*Esber and Holland, 2014*). The delay to omitted reward in a number downshift condition reduces the downshift association with the cue. In contrast, in concentration downshift conditions, rats immediately know when they have received a reduced concentration and do not exhibit behavioral excitation. We shifted the size because it is more similar to concentration in terms of when rats learn that they have received the downshifted outcome. This change will enable us to differentiate between the upshifted and downshifted cues behaviorally because the former will lead to excitation and the latter inhibition.

Recording in this setting, we replicated our prior results. We found again that many OFC neurons developed responses to the cue that predicted more reward. Interestingly, these neurons again exhibited relatively restricted responses, suggesting they were signaling something unique about this manipulation. Furthermore, they were joined by other neural populations that developed similar responses to the cue that predicted less reward and even, to some extent, to the cue that predicted no change in reward. This pattern of activity is not consistent with coding of value *independent* of outcome features and instead suggests that much of the neural activity in the OFC is tightly linked to the attributes of impending events: value being one such attribute and outcomes being one such event.

## Results

We recorded single-unit activity in the OFC in 18 rats during an odor-based unblocking task (*Figure 1a*). After implantation of microelectrodes in the lateral OFC, rats were trained to sample an

odor in a central port following house light illumination and then respond to a reward well below for a single medium-sized drop of flavored milk. This training was extensive, lasting for at least four days, and was meant to establish the initial odor as a reliable predictor of this specific outcome. Each rat then underwent 1–6 rounds of unblocking.

Each round of unblocking began with two days of training and consisted of four trial types (*Figure 1a*, compound training). One type was a reminder: the initially trained odor was followed by the expected outcome. On the other three trial types (upshift, downshift, blocked), rats were presented with the initially trained odor, followed immediately by one of three novel odors. On blocked trials, the expected medium-sized drop of milk followed the novel odor. This outcome is fully predicted by the initial odor, thus the novel odor should be blocked from acquiring associative significance. On upshift trials, a noticeably larger drop of milk followed the novel odor. Here the value is increased by the additional amount, with minimal changes in features since the reward identity is the same, as is the location and timing of delivery. Thus the novel target odor should enter into associations with the higher value and possibly with the feature of 'larger'. On downshift trials, a noticeably smaller drop of milk followed the novel odor. Here the value is decreased by the missing amount, thus the novel odor should enter into associations with the lower value and possibly with the feature of 'smaller'. We reasoned that if neurons responding to the upshifted odor cue, apparent value coding neurons in the prior study, were in fact signaling value, then they should show less firing to the blocked cue (which predicts no change in value) and even less to the downshifted cue (which predicts diminished value) (*Figure 1d*, left). On the other hand, if upshift neurons were signaling the attribute of 'larger', then we might expect them to be tuned to this particular cue, with additional populations showing selectivity to the cues predicting 'smaller' and 'no change' (*Figure 1d*, right).

## Rats learned to respond differently to the upshifted and downshifted cues

In the unblocking sessions, rats were sensitive to presentation of the novel odors, exhibiting longer latencies to respond at the reward well following odor sampling on these three trial types. Longer latencies to the novel odors were most apparent on the very first trial of each session, particularly on day 1. ANOVA revealed a main effect of trial ($F_{29,14100}$ =4.44, p<0.001) and a trial x day interaction ($F_{29,14100}$ =1.57, p<0.05). In addition to this effect, the rats also learned that two of these odors predicted meaningful changes in the outcome were meaningful. This was evident in the extinction probe test in which they initially spent more time in the fluid well following sampling of the upshift odor and less time following sampling of the downshift odor, versus the blocked odor, as if expecting more and less reward respectively on up- and downshift trials (*Figure 1b*).

## Upshift-responsive neurons in the OFC do not signal value

We recorded 334 single units during the first day of unblocking and 346 units on the second unblocking day in 60 rounds of training across all 18 rats (*Figure 1c*). Characteristically, cells in these populations fired to all of the events that characterized the trials in our task, including prominently during odor sampling (*Figure 1e–g*, steepest portion of slope). To address our hypothesis, units with a baseline firing rate below 10 Hz from both unblocking days were screened for phasic responses to one of the four odors using a t-test, which compared firing rates during the ITI and novel odor period (significance level = p<0.01). Excluding non-associative neurons, which fired from the start of training to all of the odor cues (see *Figure 2—figure supplement 1* for this and other response types not the focus of this paper and *Figure 2—figure supplement 2* for activity of all units with a baseline firing rate below 10 Hz regardless of odor response), this screen identified 120 units (Day 1=65, Day 2=55) that showed a significant increase in firing to at least one of the odor cues but not all four odors similarly.

Our primary goal in the current experiment was to test whether upshift-responsive neurons, identified in our prior study, exhibit a firing pattern consistent with value signaling, or whether they might be signaling features. Thus we focused our initial analysis on the neurons (60/120) that showed a significant phasic response to the upshifted odor. For this analysis, we compared firing to the three novel odor cues via pairwise t-tests between the baseline firing rate in the 2 s prior to the light cue and the firing rate in the 1 s following novel odor onset (p<0.01). We isolated cells that responded significantly above baseline to the upshift odor but not all four odors similarly. Cells that responded

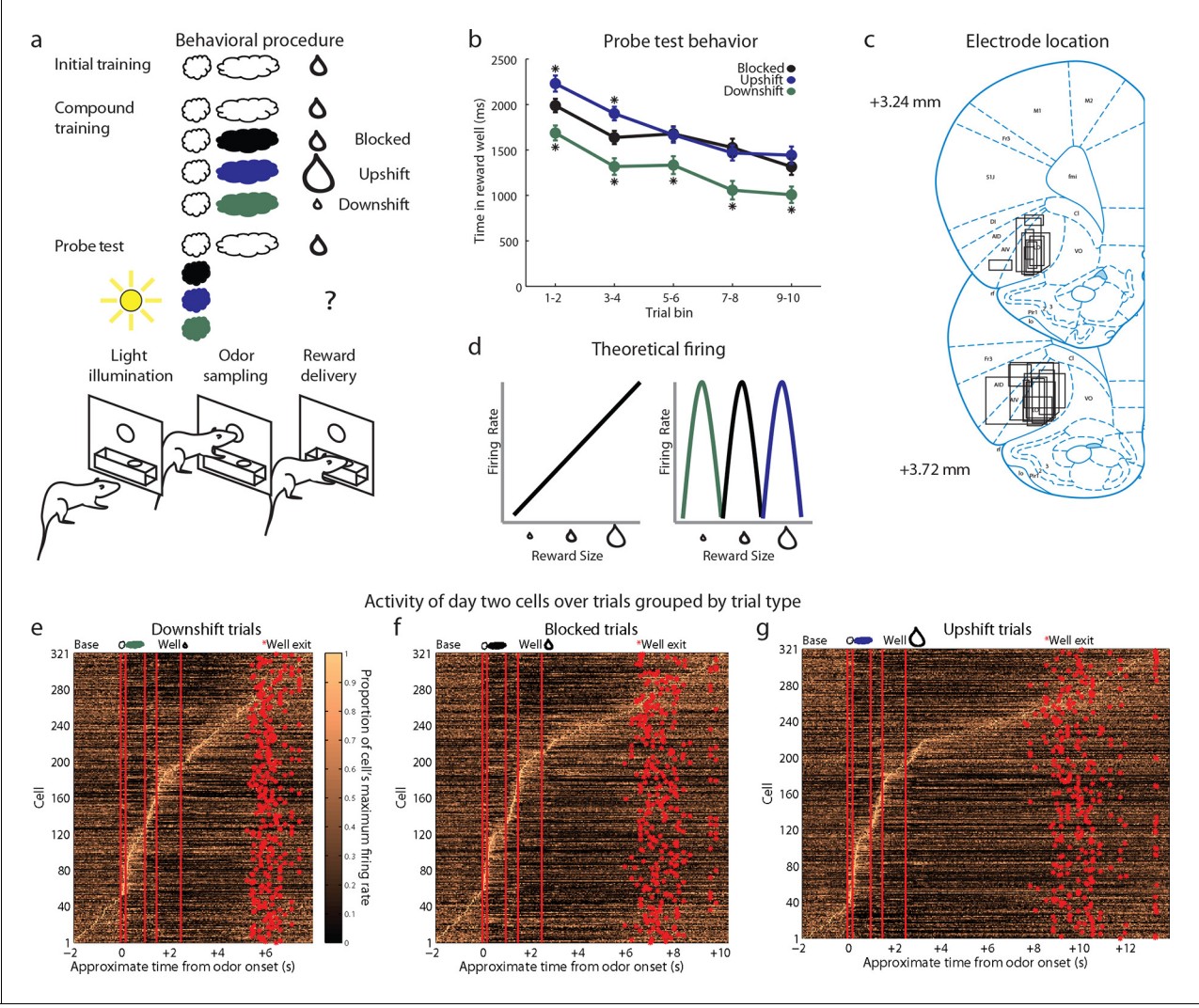

**Figure 1.** Experimental outline, behavior summary, recording sites, and recorded cells. (a) Thirsty rats were initially trained to enter an odor port after a house light lit up, then to go to the reward well below to receive a drop of chocolate milk. There were 4 trial types in the unblocking session. The first was a reminder of initial training. On the other three trial types, the originally trained odor was briefly presented, followed by one of three novel odors. The reward following the novel odors was either unchanged (black; blocked trials), larger in size (blue; upshift trials), or smaller in size (green; downshift trials). In the probe test stage, we assessed learning by presenting the novel odors without a subsequent reward. (b) Time in the reward well on the probe test trials. ANOVA for time spent in the reward well with odor (blocked, upshift, downshift), and trial (1–10) as factors found a significant effect of odor (ANOVA, $F_{2,1770} = 37.8$, $p<0.001$) and trial (ANOVA, $F_{9,1770} = 15.7$, $p<0.001$). Planned comparisons confirmed that in the first four-trial block, rats spent significantly more time in the reward well following the upshift odor ($p<0.01$) relative to the blocked odor. Rats also spent less time in the reward well following the downshift odor on all trials relative to the blocked odor ($p<0.01$). *$p<0.01$. Error bars indicate SEM. (c) Single unit activity was recorded from the lateral orbital and agranular insular cortices. Locations are shown at 3.24 and 3.72 mm anterior to bregma. AIV, AID = agranular insular area, LO = lateral orbital cortex. (d) Theoretical firing pattern for cells that signal value monotonically (left) or are tuned to individual reward sizes (right). (e–g) Activity of day two cells with a baseline firing rate under 10 Hz sorted by when each cell reaches its' maximal firing rate over the course of a trial for e) Downshift, f) Blocked, g) and Upshift trials. Cells are sorted independently on each trial type and thus the cell numbers do not correspond between trial types.

to all four odors and were value-coding were included in the upshift-responsive population. This analysis revealed a diverse set of patterns within the upshift-responsive population. While these neurons exhibited a variety of patterns of activity to the other odor cues (*Figure 2a–d*), as a group they fired most strongly to the upshifted cue (*Figure 2e*) and this preference was acquired with learning (*Figure 2f, g*), demonstrating that it was not driven by physical properties of the odors but instead by something the odor cue predicted about reward.

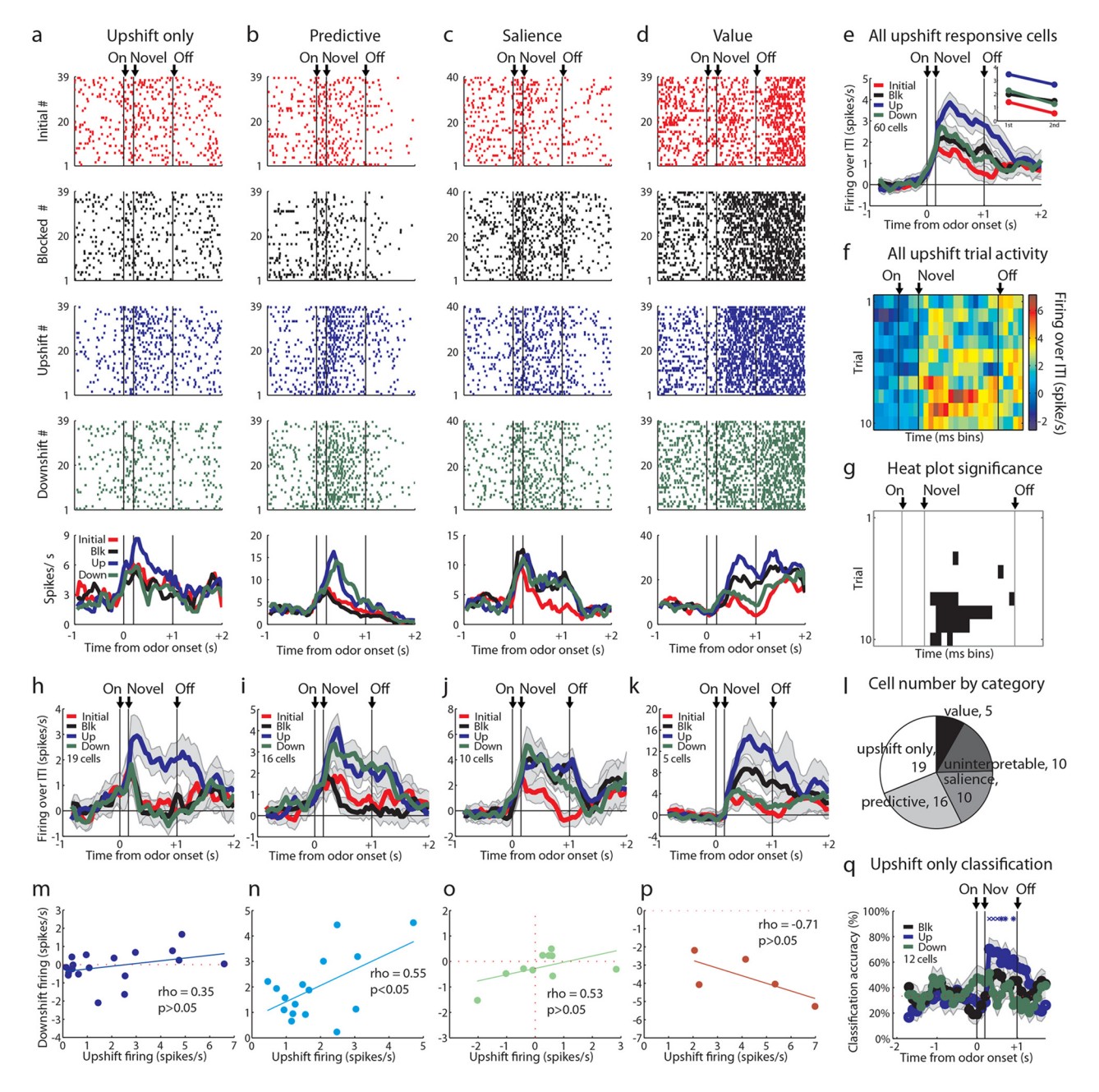

**Figure 2.** Single unit and population firing of upshift-responsive neurons. (**a–d**) Raster plots for firing of single units on initial (red), blocked (black), upshift (blue), and downshift (green) trials. All raster plots are from the first day of unblocking. Odor onset (On) is indicated by the first vertical line, novel odor onset (Novel) by the second vertical line, and odor offset (Off) by the third. Each tick represents a spike. Average activity across all trials is plotted by odor (bottom) showing **a**) Selective firing to the upshift odor, **b**) Putative predictive firing to the upshift and downshift odors, **c**) Putative salience firing to all three novel odors but not the initially trained odor, **d**) Putative value firing, exhibiting a monotonic firing pattern. (**e**) Mean neural activity (novel odor epoch - ITI) for the upshift-responsive population (n=60) is plotted. Line colors as in raster plots; shaded areas indicate standard error of mean. ANOVA with bin and odor as factors found significant effects of bin, odor, and the bin x odor interaction ($F_{59,14160}$=19.81, $F_{3, 14160}$=52.47, $F_{177, 14160}$=1.46, ps<0.001). ANOVA restricted to the novel odor period with odor and time (1st 500 ms vs. 2nd 500 ms, shown in upper right inset) as factors found a main effect of odor and time ($F_{3,472}$=13.93, $F_{1,472}$ = 10.89, ps<0.01). At both times there was no difference between blocked and downshift firing rates. (**f**) Baseline-normalized firing to the upshift cue for the upshift-responsive population on the first 10 trials of the first unblocking day (n = 33). Firing was calculated in a 150 ms sliding window for each 50-ms bin moving away from novel odor onset. The firing rate over ITI was then plotted, with dark red bins indicating maximal firing to the upshift cue and blue indicating minimal firing. Heat plot values are shown on right of heat plot. (**g**) The significance of the increased firing to the upshift cue was determined by performing a one-tailed t-test, comparing increases in firing to 0,

*Figure 2 continued on next page*

*Figure 2 continued*

using a significance of p<0.001 and a sliding window as in *Figure 2f*. Black bins indicate significant elevations in firing to the upshift cue. (h–k) Mean neural activity (novel odor epoch - ITI) for different upshift-responsive populations is plotted as in *Figure 2d* for h) Cells that respond only to the upshift cue over baseline, i) Cells that respond to the upshift and downshift cues over baseline but do not respond to the blocked cue. j) Cells that respond to the novel odor cues but not the initially trained cue over baseline. k) Cells whose signal increases monotonically with increasing reward amount. (l) Pie chart of the proportion of upshift-responsive neurons in each category. (m–p) Scatter plots of blocked-cue and baseline-normalized upshift and downshift firing rate for m) Upshift only n) Predictive o) Salience p) Value populations. (q) Classification accuracy for all cells that respond only to the upshift cue on day 1. Chance is indicated with a dashed red line. Classification accuracy significance above chance is indicated above time bins in a color matching the trial type. *p<0.05, ˣp<0.01. Error bars indicate SEM.

The following figure supplements are available for figure 2:

**Figure supplement 1.** Odor-responsive units not analyzed in the main text.

**Figure supplement 2.** Recorded units not restricted to odor-responsive units.

However, firing across the population did not appear to provide the linear value correlate predicted by a value-coding hypothesis (*Figure 1d*, left). This was evident in the population response, which while highest for the upshifted cue, did not distinguish between the blocked and downshifted cues (*Figure 2e*). Of course, the heterogeneity of response types within such a broad population could obscure a simple value correlate, so next we asked whether a subpopulation of the 60 upshift-responsive neurons might show a linear value correlate. Value-coding was operationally defined as an excitatory response to at least one odor and significantly more activity to the upshift cue than the blocked cue as well as significantly less activity to the downshift cue than the blocked cue. Few neurons exhibited firing that clearly rank ordered the cues according to their value (*Figure 2k*; 5/60; 8%). Though firing in these neurons did provide a linear value correlate (*Figure 2p*), this proportion was the smallest of the theoretically meaningful patterns that we found and its size was not larger than chance given our analytical approach. We repeated this analysis on all cells regardless of baseline firing rate and without limiting our analysis to odor-responsive cells and found a total of 6 cells, suggesting that our screen had not prevented us from finding value-coding cells.

On the other hand, we found many neurons that showed isolated firing to the upshifted odor alone (*Figure 2h*; 19/60; 32%). Indeed this was our largest subgroup. In these cells, we were only able to accurately classify the upshift trial type above chance, and no other trial type, suggesting that these cells only contain information about the upshift cue (*Figure 2q*). Additionally we saw an almost equally large population of neurons that fired similarly to the upshifted and downshifted cues (*Figure 2i*; 16/60; 27%). This population was reminiscent of the neurons in our prior study that fired similarly to the two cues predicting changes in the outcome. And finally we found a substantial number of upshift-responsive neurons that were equally responsive to the other two novel odor cues (*Figure 2j*; 10/60; 17%), replicating the finding of salience or novelty encoding in our prior study. Notably, none of these populations showed a relationship between firing to the upshift and downshift cues consistent with value coding across them (*Figures 2m-o*). Thus, if we disregard the very small number of neurons showing linear coding of value, our results here largely replicated what we observed when unblocking learning by shifting value versus identity (*Figure 2l*). Reducing the feature space by moving to a single bolus of reward and including an explicit bidirectional value shift did not reveal frank value coding in a large proportion of the neurons.

## OFC neurons respond independently to the upshifted, downshifted, and blocked cues

The upshift-responsive population may respond in a non-linear fashion to value. It is not immediately clear why that would be, since rats have been exposed to the different valued rewards in the session, and thus firing in the OFC should scale to reflect the full range that is available (*Padoa-Schioppa, 2009*). While this is difficult to rule out, one way to address this is to turn to our alternative account, which is that the firing of these neurons represents the attribute or feature that is predicted by the cue, which is the increase in the size of the reward. If this is what is signaled by these neurons, then one would expect there to be similar populations firing to the cue predicting a

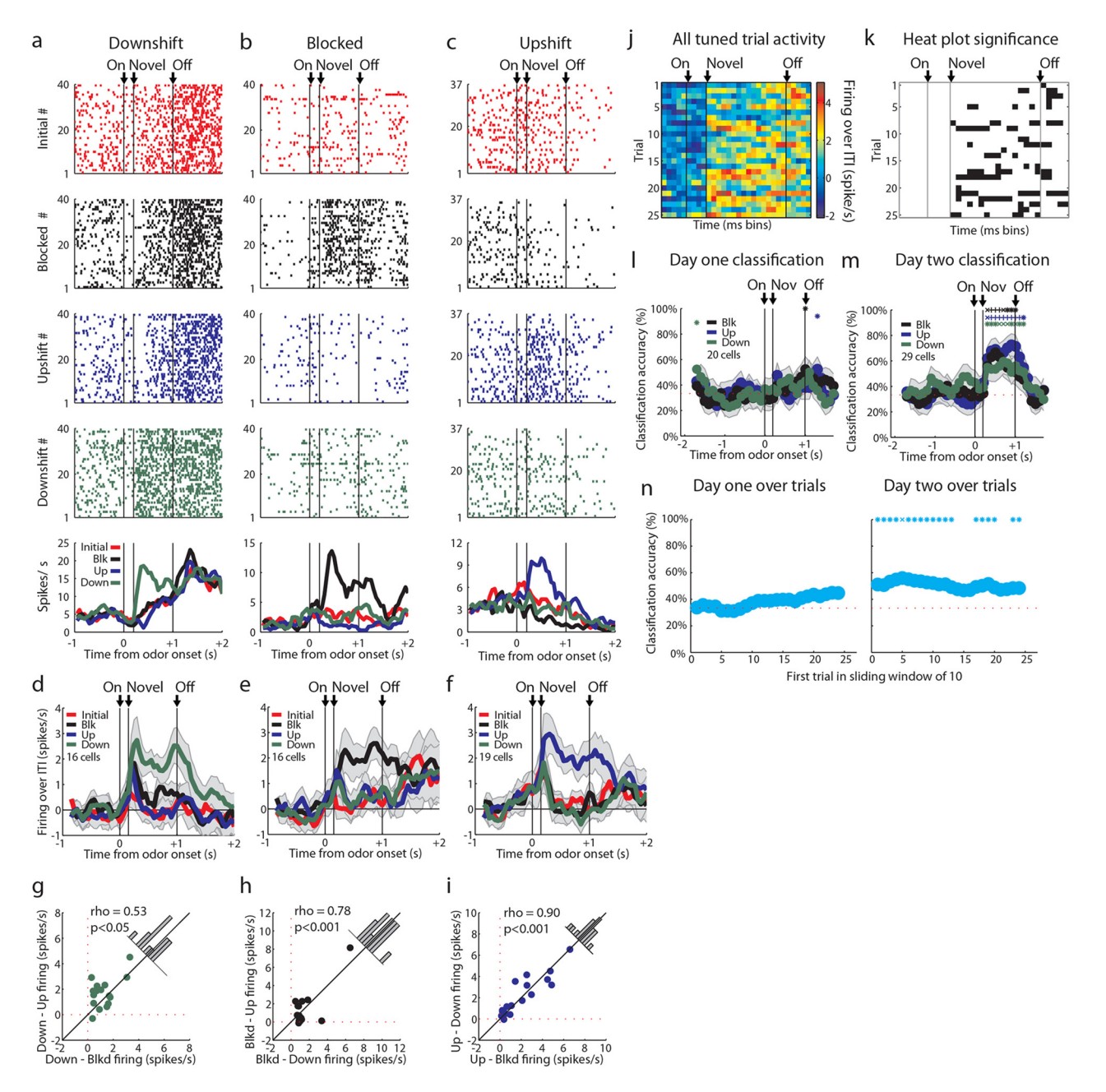

**Figure 3.** Tuned cells and population activity. Raster plots as in *Figure 2a–d* for firing of single units that are selective for one cue on unblocking day 2 are shown for units responsive to only the (**a**) Downshift cue (**b**) Blocked cue (**c**) Upshift cue. (**d-f**) Mean neural activity (novel odor epoch - ITI) as in *Figure 2e* for the d) Downshift, e) Blocked, f) Upshift responsive populations. (**g-i**) Scatter plots in which firing for each non-preferred cue is subtracted from the preferred cue for tuned populations. Bar graphs show the distribution of difference between the two indices for each neuron. To the extent that cells do not differentiate between the two non-preferred cues, scatter points should congregate around the diagonal, the histogram bars should peak in the center, and a t-test should indicate that the distribution of the mean is not significantly different from 0. **g**) Scatter plots and histograms are shown for the downshift population. A t-test of the diagonal distribution data found that the distribution of the mean is not significantly different from 0, p=0.07. **h**) Scatter plots and histograms are shown for the blocked population. A t-test of the diagonal distribution data found that the distribution of the mean is not significantly different from 0, p=0.80. **i**) Scatter plots and histograms are shown for the upshift population. A t-test of the diagonal distribution data found that the distribution of the mean is not significantly different from 0, p=0.63. (**j-k**) Heat plots and p-value plots as in *Figure 2 f-g, p*<0.01, for all tuned cells on unblocking days 1 and 2 (n = 51) normalized to the initially trained cue. (**l-m**) Classification accuracy over time from odor onset as in *Figure 2q* for l) 20 tuned cells on unblocking day 1 and m) 29 tuned cells on unblocking day 2. Classification accuracy significance above chance is indicated above time bins in a color matching the trial type. *p<0.05, *p<0.01., +p<0.001. (**n**) Classification accuracy for tuned cells over trials

*Figure 3 continued on next page*

*Figure 3 continued*

on day one (n=20) and day two (n=29) over a sliding window of 10 trials averaged by trial type. Classification accuracy significance above chance is indicated above time bins. *p<0.05, *p<0.01. On day one, accuracy does not rise above chance, but there is a correlation between trial number and classification accuracy, rho = 0.85, p<0.001. Error bars indicate SEM.
The following figure supplement is available for figure 3:

**Figure supplement 1.** Tuned cells' trial firing and latency to reward well.

decrease in reward size and perhaps to the cue signaling no change. These neurons would not have passed our first screen, since we specifically searched for upshift-responsive neurons. So to capture these, we returned to our original data set and repeated the analysis above, but this time examining firing to the other cues. This approach found cells that exhibited isolated firing to the downshifted cue (n = 16, *Figure 3a, d, g*) and to the blocked cue (n = 16, *Figure 3b, e, h*). These populations were similar to the neurons that showed isolated firing to the upshift cue (n = 19, *Figure 3c, f, i*). The difference in the number of cells that responded to each cue was not significant, chi-squared = 0.353, p = 0.84. In each case, cue-evoked firing developed with training (*Figure 3j, k*). This impression was confirmed by an ensemble analysis that tested how well these neurons could identify the trial type; this analysis found that classification accuracy by these cells during the odor-sampling period improved significantly from day 1 (n = 20, *Figure 3l*) to day 2 (n = 29, *Figure 3m*). Within day 1, classification accuracy was at chance during the odor-sampling period. There was significant correlation between trial number and classification (rho = 0.85, p<0.001) but accuracy did not increase to above chance until day 2, beginning on the first ten-trial window shown and remaining above chance for the majority of trial windows. The chance decoding in the first day reflects the finding that the responses of these neurons were not stable across these trials (*Figure 3n*). Further, high decoding accuracy on the second day suggests that a downstream structure could decode the trial type from these tuned cells. Though the magnitude of the change in firing rate between baseline and the preferred cue is ~2 spikes/second in tuned cells, the high decoding accuracy suggests that the variance in firing plays a significant role in downstream decoding. High decoding accuracy on day 2 suggests that there is low variance in this response, allowing a downstream structure to interpret a relatively small change in firing magnitude.

Finally we also examined whether firing activity in these populations might reflect value or at least vigor of responding to that specific cue. For this we compared firing in each neuron in each population to the speed of responding on those trials and also on trials involving the other two cues. Although a handful of individual neurons showed a significant correlation with response latency, there was no systematic relationship ( *Figure 3—figure supplement 1*).

## Value coding in OFC neurons is not 'blocked' by our task

We conducted our original study to isolate the role of OFC in representing valueless information about predicted outcomes. We used blocking and identity unblocking as a way to prevent information about general value from becoming associated with our cue of interest. We found that OFC neurons developed firing to a cue predicting a valueless change in reward identity. However, many OFC neurons also fired when we increased the amount of reward. Here we conducted a follow-up experiment designed to test whether these responses are signaling value or whether they might be signaling the feature of the larger reward. We found primarily the latter, with very few neurons showing linear value correlates.

However, one flaw in our design is that, of course, we intentionally used blocking to reduce or eliminate associations with general value. Our apparent failure to observe linear value correlates could be secondary to this manipulation. To address this, we ran a control experiment in which the trial structure was identical to the above experiment except that in compound training, rats did not receive training with novel cues in compound with the initial odor cue (*Figure 4a*). This change essentially eliminated the blocking of value, without changing any of the other aspects of the task. Accordingly, we found that the rats again showed evidence of learning about the reward predicted by the three cues, spending more time in the fluid well following sampling of the large-predictive

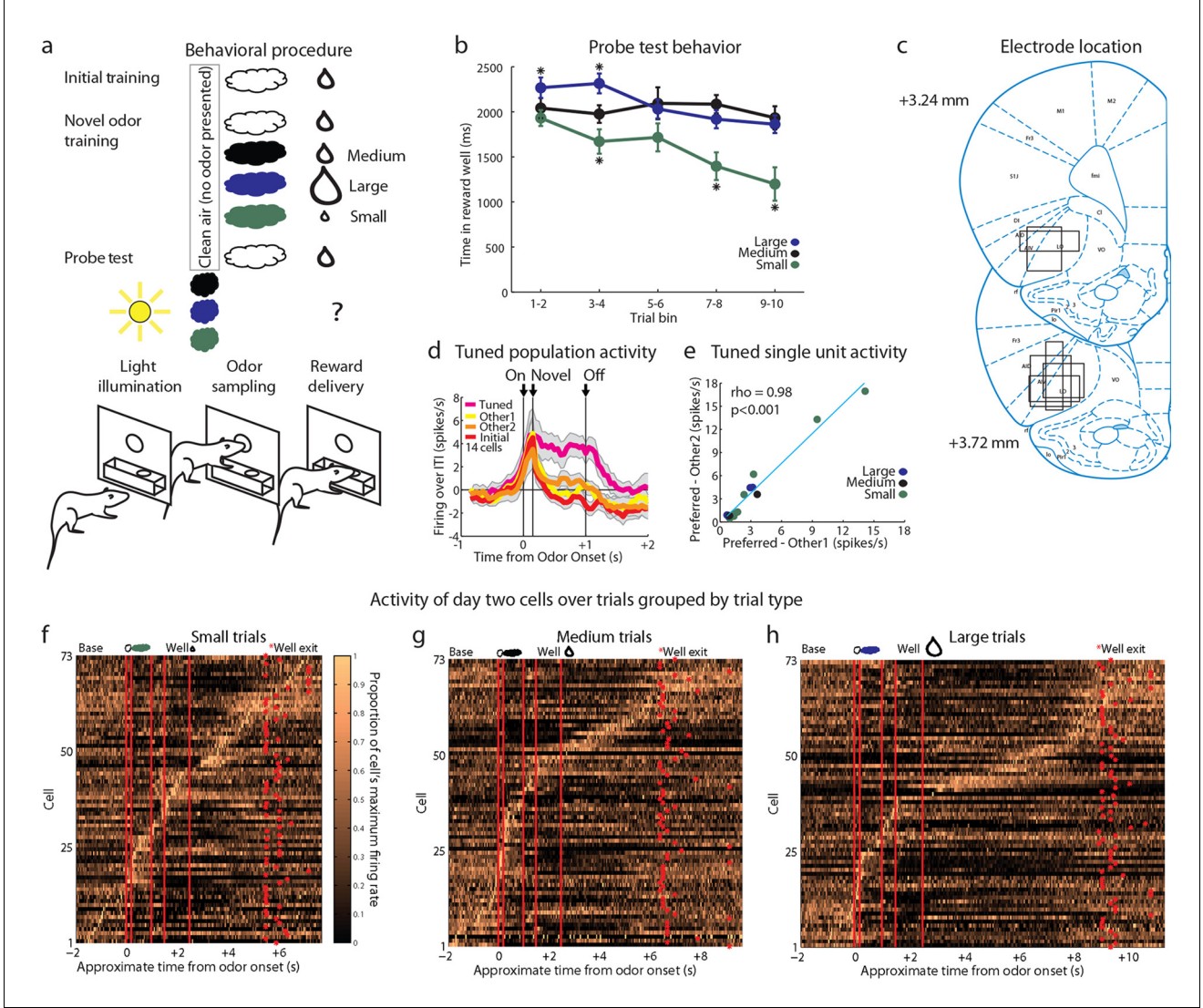

**Figure 4.** Control experimental outline, behavior summary, recording sites, and neural results. (a) Procedure is nearly identical to that in *Figure 1a* except a control procedure was used rather than a blocking procedure. The reward following the novel odors was either medium (black), large (blue), or small (green). (b) Time in the reward well on the probe test trials. ANOVA for time spent in the reward well with odor (small, medium, large), and trial (1–10) as factors found a significant effect of odor (ANOVA, $F_{2,450} = 22.61$, $p<0.001$) and trial (ANOVA, $F_{9,450} = 2.7$, $p<0.01$). Planned comparisons confirmed that in the first four-trial block, rats spent significantly more time in the reward well following the large odor ($p<0.05$) relative to the medium odor. Rats also spent less time in the reward well following the small odor on all trials relative to the medium odor ($p<0.001$). *$p<0.05$ (c) Single unit activity was recorded from the lateral orbital and agranular insular cortices. Locations are shown at 3.24 and 3.72 mm anterior to bregma. AIV, AID = agranular insular area, LO = lateral orbital cortex. (d) Mean neural activity (novel odor epoch - ITI) is plotted for all of the tuned cells (n=14) combined, with the tuned cue in pink and the initially trained cue in red. Other 1 = medium for small-responsive and large-responsive and small for medium-responsive. Other 2 = large for small-responsive and medium-responsive and small for large-responsive. Error bars indicate SEM. (e) Scatter plot as in *Figure 3g-I* for all tuned cells. Other 1 = medium for small-responsive and large-responsive and small for medium-responsive. Other 2 = large for small-responsive and medium-responsive and small for large-responsive. (f-h) Activity of day two cells as in *Figure 1e-g* for f) Small, g) Medium, h) Large trials.

odor and less time following sampling of the small-predictive odor, versus the medium-predictive cue, as if expecting more and less reward respectively after sampling these cues (*Figure 4b*).

Against this backdrop, we then asked whether this modified design had uncovered the linear value coding that seemed to be missing in our unblocking design. We recorded 93 single units during the first day and 78 units on the second day in 16 rounds of training across all 7 rats (*Figure 4c*). As in our first experiment, these neurons exhibited firing to all of the events that described a trial,

including prominently to the odor-sampling period (*Figures 4f–h*). Twenty-six units (Day 1=16, Day 2=10) showed a significant increase in firing to at least one but not all four of the odor cues similarly. We found no evidence of linear value coding (0/26, 0%); instead neurons in this population again tended to develop firing that was tuned to the individual cues (14/26, 54% including 8 small, 2 medium, and 4 large). Although we had a much smaller sample, the activity of these neurons looked similar to the activity of the tuned populations identified in our first experiment inasmuch as these neurons exhibited high firing to one of the odors and equally low firing to the other two (*Figures 4d, e*).

## Discussion

Neural signals in the OFC are often described as representing either outcome expectancies or abstract value. Although many studies have argued for one or the other, few have used behavioral designs that clearly dissociate predictions of these two hypotheses. Previously we addressed this question by using an unblocking procedure to strip away or 'block' the abstract value of the outcome during learning, while leaving unblocked – free to enter into associations – the outcome's sensory and other unique features. This approach revealed that many neurons in the OFC developed firing to target cues that predicted a valueless change in the identity of the predicted outcome. This sort of associative encoding is consistent with the involvement of the orbitofrontal cortex in a variety of behaviors that require direct access to specific information about the outcome's sensory features (*McDannald et al., 2005*; *2011*; *Ostlund and Balleine, 2007*; *Howard et al., 2015*).

However, we also found that many neurons developed firing to a cue that simply predicted the delivery of an additional drop of reward. Although this condition was included as a positive control condition (in the event we saw no change in activity to our other cue), we were intrigued because the firing to this cue was not substantially stronger or in any other way more robust than that to the cue that predicted an unambiguous change in reward identity. This seemed at odds with proposals that the primary or at least a major role of the OFC is to encode the abstract or general value that cues predicting reward acquire (i.e., independent of outcome features). We wondered whether firing to this cue was simply reflecting some unique feature of additional reward (the concept of more, its timing, the unique presence of a third drop) and not the abstract increase in value it represented.

Here we tested this hypothesis by repeating our prior study, to identify neurons that fired to the cue predicting more reward, while adding a condition in which a cue predicted less reward. We reasoned that if the neurons firing to more reward were signaling abstract value, independent of any features unique to that condition, then neurons that fire more to a cue predicting more reward should fire less to a cue predicting less reward (versus the cue predicting no change). On the other hand, if these neurons were signaling a unique feature of the larger reward - essentially something contributing to its identity - then these neurons would not become selective for the other cues.

Our results were clearly in line with the latter prediction. We saw only a small population of neurons that exhibited firing consistent with abstract value coding across the three cues (upshift, downshift, blocked) and much larger groups of neurons that became responsive to each cue with training. Notably this was true in both the first experiment, in which we used a pre-trained 'blocking' cue as we had in the first study, and it was also true in the second experiment, in which we eliminated this pre-training. Thus, we do not believe that our failure to observe abstract or general value coding across cues was due to blocking of value by the pre-trained cue.

The firing that develops to these cues may reflect associations with novel features of the outcome delivered on these trial types (larger, smaller, unchanged). Such associations are known to develop when additional rewards are delivered, even during unblocking (*Holland, 1984*). This information encoded across an ensemble of neurons would clearly be relevant to determining the value of the impending outcome. However, our findings are not consistent with the proposal that these neurons directly signal the abstract or linear value that is acquired across cues at the single unit level. Indeed we saw almost no neurons in the OFC that fit this profile in our experiments.

So what does our failure to find these correlates mean? Many prior studies have reported value coding in the OFC (*Plassmann et al., 2007*; *Padoa-Schioppa and Assad, 2006*; *Levy and Glimcher, 2011*; *Padoa-Schioppa, 2009*; *Tremblay and Schultz, 1999*; *Padoa-Schioppa and Assad, 2008*; *Padoa-Schioppa, 2013*; *Cai and Padoa-Schioppa, 2014*; *McNamee et al., 2013*; *Hare et al., 2008*; *2010*; *Plassmann et al., 2010*; *Kahnt et al., 2014*; *Strait et al., 2014*). How do we reconcile our

results with these prior reports? A number of possible ways exist to understand this apparent discrepancy. The first is that it represents a species difference. Most of the aforementioned studies have focused on 'economic' value and decision-making. These models are largely applied to humans and monkeys. It is possible that the increased size and complexity of the OFC in primate species has led to the development of this abstract value coding. This is possible, though we believe it is unlikely, since most of the anatomical and functional data across species is roughly similar (*Rudebeck and Murray, 2014*; *Wallis, 2012*; *Stalnaker, et al., 2015*). This is especially true of the importance of the OFC to reinforcer devaluation (*Gottfried, 2003*; *Gallagher et al., 1999*; *Izquierdo et al., 2004*), a function first identified in rodents. To the extent economic value is sensitive to devaluation (*Padoa-Schioppa, 2011*; *Padoa-Schioppa and Schoenbaum, 2015*), this suggests that the role of the OFC in contributing to its representation is not unique to primates.

A second possibility relating to species differences is the accompanying differences in task design. Studies claiming value effects on neural activity in OFC have typically tested in a situation requiring choices between outcomes. This may make value information particularly salient compared to our task where explicit comparisons are not required. We view this as unlikely, since a core requirement of value signaling is automaticity (*Lebreton et al., 2015*). Nevertheless, this does not preclude differences in how this tracked value is represented in neural activity in the OFC. While the OFC may perform the same function between Pavlovian and choice tasks, there may be differences in how each is encoded.

A third possibility is that we are in the wrong part of the OFC to find these correlates. We are recording primarily in the lateral OFC. This is an area distinguished by a pattern of connectivity with amygdala, striatal and cortical areas that is qualitatively most similar to what is termed the lateral orbital network in primates (*Ongur, 2000*; *Price, 2007*). Most of the studies that have identified abstract value correlates in the OFC found them in more medial areas (*Plassmann et al., 2007*; *2010*; *Levy and Glimcher, 2011*; *McNamee et al., 2013*; *Hare et al., 2008*; *2010*; *Strait et al., 2014*). Further, studies in primates that have tried to distinguish medial versus lateral functions have assigned valuation based functions more to medial areas and functions such as credit assignment and representation of reward identity to more lateral areas (*Noonan et al., 2010*; *2012*). It may be that the sort of linear value correlates that we were looking for are to be found in medial OFC. Indeed recent work in humans has shown that OFC represents specific outcome features and that more lateral orbital areas represent those outcomes in a way that is dependent upon prior cues (*Klein-Flugge et al., 2013*). In this regard, it is worth noting that the lateral part of the OFC is not normally necessary for behaviors in which value independent of outcome features is required. For example, the OFC is not required for simple Pavlovian or instrumental conditioning (*Ostlund and Balleine, 2007*; *Gallagher et al., 1999*; *Gremel and Costa, 2013*; *Izquierdo et al., 2004*) discrimination learning (*Izquierdo et al., 2004*; *Walton et al., 2010*; *McDannald et al., 2005*), extinction by reward omission (*Takahashi et al., 2009*), transfer (*Ostlund and Balleine, 2007*), and even perhaps reversal learning (*Rudebeck et al., 2013*), all of which can be accomplished without reference to specific information about predicted outcomes. Similarly, both blocking and unblocking – when it can be accounted for by value – do not require the OFC (*Burke et al., 2008*; *McDannald et al., 2011*). However, the OFC is necessary for superficially similar behaviors (Pavlovian or instrumental responding, discriminations, even learning) when they require knowledge of the outcome features in order to recognize errors or to derive or infer a value (*Ostlund and Balleine, 2007*; *Gallagher et al., 1999*; *Gremel and Costa, 2013*; *Izquierdo et al., 2004*; *McDannald et al., 2011*). This is even true in unblocking, where we have shown that the OFC is required for the development of conditioned responding to a target cue paired with a shift in outcome identity but not to a target cue paired with additional outcome (*McDannald et al., 2011*).

A fourth possible explanation for the apparent discrepancy between our results and many of these studies lies in the difference in measures. The vast majority of these studies have reported that BOLD signal correlates with abstract value (*Plassmann et al., 2007*; *2010*; *Levy and Glimcher, 2011*; *McNamee et al., 2013*; *Hare et al., 2008*; *2010*). BOLD signal differs from single unit spiking in at least three important ways that are relevant to this question. First, BOLD signal obviously reflects the summed activity of many neurons. Thus it identifies what the global summed processing across a large ensemble is tracking. This might be very different from what is encoded by groups of individual single units. Without knowing more about specific connectivity between neurons within an area, it is impossible to know if one or the other result is more relevant to the function of the area.

But at a minimum, this might lead to very different results. The second way these two measures differ is that BOLD signal is more likely to reflect neural processing other than what is output from an area. This is because it reflects energy usage, which is a measure that does not distinguish between local and long distance communication. On the other hand, extracellular recording electrodes are heavily biased to record from large, regular spiking neurons (*McCormick et al., 1985*), which are likely to be the output neurons, at least of cortical regions. Again this might cause very different signals to be detected using the two methods. Finally the third and related difference between these two measures is that fMRI can be heavily influenced by input to an area (*Logothetis et al., 2001*; *Logothetis and Pfeuffer, 2004*; *Logothetis and Wandell, 2004*; *Logothetis, 2008*). Again this is because it measures energy usage and thus will be sensitive to the EPSP's (and IPSP's) due to afferent input. Note this is true even if these inputs do not lead to summation and action potential generation at the downstream hillock of a large output neuron (a very energy efficient event). For all these reasons, it is to be expected that fMRI studies and single unit studies may often report divergent findings.

Finally it is worth noting that neuroeconomic value may not be the same sort of value identified in our lexicon as pure, abstract, general or cached (i.e., independent of outcome features). This has been our reading of the original reports (*Padoa-Schioppa and Assad, 2006*; *Levy and Glimcher, 2011*; *Plassmann et al., 2007*). However one of the authors of these studies has suggested more recently that neuroeconomic value is in fact tied closely to and derived from knowledge of the features of the predicted outcome (*Padoa-Schioppa and Schoenbaum, 2015*). He states unequivocally that the economic value that correlates with firing in the OFC in monkeys is the same value that changes with devaluation and that it is tied inextricably to the identity of the predicted outcome. Indeed devaluation-sensitivity is taken to be an iconic feature of decisions based on economic value (*Padoa-Schioppa, 2011*). Notably, this definition would align our concept of signaling of outcome features with signaling of economic value, since economic value would then be just another attribute of the outcome. Of course, we would not expect it to be shared across outcomes or even necessarily across cues, particularly in a very lightly trained animal such as that in our current study. In this context, we would predict relatively specific representations and relatively little representation of this value across outcomes. This is what we have observed here. It is possible with extended training, such as that in studies reporting large populations of single unit coding value (*Padoa-Schioppa and Assad, 2006*; *2008*; *Padoa-Schioppa, 2009*; *2013*; *Cai and Padoa-Schioppa, 2014*), that the representations generalize as the rewards used become part of a 'goods space' as it is defined in these prior studies. As long as what is being signaled by such activity remains an attribute or feature of the predicted outcome - and sensitive to changes in that feature without further learning - then this would be consistent with our current data.

## Materials and methods

### Subjects

18 and 7 Male Long-Evans rats were obtained at 200–250 g from Charles River Labs, (Wilmington, MA) for the blocking and control experiments, respectively. Rats were tested at the NIDA-IRP in accordance with NIH guidelines.

### Surgery and histology

Using aseptic, stereotaxic surgical methods, a drivable bundle of sixteen 25 μm diameter FeNiCr wires (Stablohm 675, California Fine Wire, Grover Beach, CA) was chronically implanted in the left hemisphere at OFC at 3.0 mm anterior to bregma, 3.2 mm laterally, and 3.9 mm ventral to each rat's brain surface. These wires were cut at an angle with surgical scissors immediately prior to implantation, to extend ~1.8-2.5 mm beyond the cannula, with a range of ~0.3 mm between wires. Current was passed through each electrode immediately prior to implantation to lower the impedance to ~300–400 kOhms. At the study's conclusion, a 15 μA current was passed through each electrode to mark the final position. Following perfusion of the rats, their brains were extracted and processed for histology using standard techniques.

## Blocking task

Recording was conducted in grounded aluminum chambers approximately 18'' on each side with sloping walls narrowing to an area of 12'' x 12'' at the bottom. An odor port was located centrally above a fluid well on a panel in the right wall of each chamber. Above the panel were two lights. To allow rapid delivery of olfactory cues to the odor port, it was connected to an airflow dilution olfactometer. Odors were chosen from compounds obtained from International Flavors and Fragrances (New York, NY). The fluid well was connected to lines controlling the independent delivery of liquid rewards. A computer running a behavioral program written in C++ implemented control of the task. Following implantation with microelectrodes, rats were water deprived by restricting access to 10 min daily. Following two days of water deprivation, rats were shaped, in stages, to hold in the odor port for 1 s in order to receive a water reward at the well. Each trial started with house light illumination, following which rats had 3 s to enter the odor port. A failure to enter the odor port caused restart of the trial. Rats were required to hold for 1 s in the odor port, and upon exit had 3 s to enter the reward well. Again, failure to hold for 1 s or to make reward well entry within 3 s resulted in restart of the trial. Following shaping, rats were trained until they proficiently responded for the initial odor to receive a medium-sized bolus of a flavored milk solution; this comprised up to 15 sessions, with a maximum of 170 trials per session. Completion of ~150 trials per session was characterized as proficient responding.

Once rats were deemed proficient at initial training and single units were isolated, the unblocking procedure began. On each of the two learning days, rats received four trial types. The first trial type was a reminder of initial training. The remaining trial types comprised a 200 ms presentation of the initial odor followed by one of three 800 ms, novel, differentiable odors: one signaling the same medium-sized bolus of flavored milk used in prior training, a second signaling a bolus more than twice as large, and a third signaling a bolus less than half the size of the medium bolus. The behavioral requirements for each of trial type were exactly as in initial training. Rats completed 20–40 trials with each novel odor per session during unblocking. Then, on the probe test day, rats received 10 reminder trials of each type, followed by up to 10 trials of each novel odor alone without reward, interleaved with rewarded presentations of the initial odor to maintain responding. During the unrewarded, novel-odor extinction trials, both requirements to sample the odor for 1-s and respond to the reward well were lifted. This unblocking procedure was repeated one to six times per rat, using a new set of blocked, upshift and downshift odors each time; for rats who completed more than 4 sessions, a new initial odor was also trained for 4–5 sessions prior to repeating unblocking.

## Control task

The Control task was identical to the Blocking task through shaping. Following shaping, rats were trained until they responded proficiently for an odor cue to receive a medium-sized bolus of water; this comprised up to 15 sessions, with a maximum of 170 trials per session. Completion criteria and instrumental components were identical to the Blocking task aside from one difference, which was that the rats were required to hold for 1 s in the odor port, but the first 200 ms for every trial type were clean air, followed by 800 ms of odor cue presentation. Once rats were deemed proficient at initial training and single units were isolated, rats were trained that a new odor predicted a medium-sized drop of a flavored milk solution. Following this, on each of the two learning days, rats received four trial types. The first trial type was a reminder of initial training. The remaining trial types comprised three novel and differentiable odors, presented as in the Blocking task, again with one signaling the same medium-sized bolus of flavored milk used in prior training, a second signaling a larger bolus, and a third signaling a smaller bolus. The behavioral requirements and probe test were identical to the Blocking task. This control procedure was repeated one to four times per rat, using a new set of small, medium and large odors each time.

## Single-unit recording

Neural activity was recorded using six identical Plexon Multichannel Acquisition Processor systems (Dallas, TX), interfaced with odor discrimination training chambers described above. Following recovery from surgery and proficiency in shaping, electrodes were advanced daily until activity was obtained. Rats received reminder training using the pre-trained initial odor, as described above, during this process. Once rats showed proficient responding and single units were isolated, the rat

began unblocking. During this three-day procedure, the electrode was moved ~167 μm between the first and second learning days for approximately three quarters of the runs in the Blocking-trained rats and for all of the runs in the Control-trained rats. Following completion of each three-day unblocking procedure and prior to repetition of this process in new odor cues, the electrode was advanced ~167 μm in order to acquire neurons in a new location in OFC in all rats.

## Statistical data analysis

Units were sorted using Offline Sorter software from Plexon Inc (Dallas, TX) using a k-means algorithm. Sorted files were next processed in Neuroexplorer to extract relevant event markers and unit timestamps. These data were then analyzed in Matlab (Natick, MA). To analyze activity in response to the novel odors, we examined activity between 300–1300 ms subsequent to initial odor onset, which corresponded approximately with the novel odor delivery to the odor port. The inter-trial interval was defined as the 2 s preceding illumination of the house light. Normalized firing was calculated by subtracting firing rate during the ITI from the period of interest: Normalized firing = (Period spikes/s) – (ITI spikes/s). Odor-responsive neurons were identified as units that showed an increase in firing from baseline during odor sampling (t-tests, $p < 0.01$) on at least one of the 4 trial types in either the Blocked or the Control experiment. All odor period analyses and figures show data beginning with the 8[th] trial, excluding heat plots over trials and classification accuracy over trials.

Neurons were classified as putative sensory neurons and excluded from further consideration if they significantly increased firing to all odors and were not value-coding. Neurons that were not eliminated by this screen were classified as upshift-responsive in the Blocked experiment if they increased firing significantly to the upshift cue. Single-unit and population activity was plotted in 50-ms bins. Population activity was additionally analyzed with repeated measures ANOVA with bin (50 ms) and odor trial (initial, blocked, upshift and downshift) as factors.

Heat plots were constructed in 150-ms sliding windows moving away from the novel odor onset in 50 ms increments. Warmer colors (dark red) indicated positive difference scores while cooler colors (dark blue) indicated negative difference scores. Significance of differential firing to identified odors was determined by performing a one-tailed t-test comparing differential firing to zero in the exact same 150-ms sliding windows for each trial shown.

Latency correlation analyses were performed between individual trial latencies and the firing rate during the novel odor period on the same trial. Correlation analyses were performed on cells recorded on the second day of unblocking on trials 5 through 30.

Classification accuracy was calculated using a linear classifier on instantaneous firing rate measured in 100 ms bins. The goal of the classification was to predict the trial type. Classification accuracy by trial was performed using a sliding window. Accuracy was calculated during the 400 ms window following novel odor presentation for all cells and 500 ms window following novel odor presentation for tuned cells. Bins were counted as additional trials to offset low trial number according to the number of cells used. Statistical significance was calculated based on a binomial distribution (*Combrisson and Jerbi, 2015*).

Single unit and population firing were smoothed by taking a four-bin average moving in 50 ms increments moving away from the novel odor period. Classification accuracy during the odor period was smoothed by taking a three-bin average moving in 100 ms increments moving away from the novel odor period. Classification accuracy over trials plots were smoothed by taking a three-trial average.

## Acknowledgements

This work was supported by the Intramural Research Program at the National Institute on Drug Abuse. The opinions expressed in this article are the authors' own and do not reflect the view of the NIH/DHHS.

## Additional information

### Funding

| Funder | Grant reference number | Author |
|---|---|---|
| National Institute on Drug Abuse | IRP | Geoffrey Schoenbaum |

The funders had no role in study design, data collection and interpretation, or the decision to submit the work for publication.

### Author contributions

NL, conceived and designed the experiment; acquired the data; analyzed and interpreted the data; drafted the manuscript; approved the submitted manuscript, Conception and design, Acquisition of data, Analysis and interpretation of data, Drafting or revising the article; MAMcD, conceived and designed the experiment; acquired the data; analyzed and interpreted the data; critically revised the manuscript; approved the submitted manuscript, Conception and design, Analysis and interpretation of data; CVS, acquired the data; critically revised the manuscript; approved the submitted manuscript, Acquisition of data, Drafting or revising the article; BFS, acquired the data; analyzed and interpreted the data; critically revised the manuscript; approved the submitted manuscript, Conception and design, Acquisition of data, Analysis and interpretation of data, Drafting or revising the article; JFC, critically revised the manuscript; approved the submitted manuscript, Conception and design, Analysis and interpretation of data, Drafting or revising the article; GS, conceived and designed the experiment; analyzed and interpreted the data; drafted the manuscript; approved the submitted manuscript, Conception and design, Analysis and interpretation of data, Drafting or revising the article

### Ethics

Animal experimentation: This study was performed in strict accordance with the recommendations in the Guide for the Care and Use of Laboratory Animals of the National Institutes of Health. All of the animals were handled according to approved institutional animal care and use committee (IACUC) protocols (#15-CNRB-108 and 12-CNRB-108) of the IRP.

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
