## [Decision Letter]

Thank you for submitting your work entitled "Lateral orbitofrontal neurons acquire responses to upshifted, downshifted, or blocked cues during unblocking" for consideration by *eLife*. Your article has been reviewed by two peer reviewers, and the evaluation has been overseen by a Reviewing Editor and Eve Marder as the Senior Editor.

The reviewers have discussed the reviews with one another and the Reviewing editor has drafted this decision to help you prepare a revised submission.

Summary:

This is an exciting follow up on a highly creative earlier study that further advances our knowledge of what drives neurons in the orbital prefrontal cortex.

Essential revisions are outlined below in detail:

*Reviewer #1:*

This manuscript is a follow-up on an earlier study investigating what drives activity in orbitofrontal cortex neurons. The task is similar to before, except that the situation where the reward identity is changed ("identity unblocking") is replaced by a condition where the reward size is reduced ("downwards unblocking"). There was again a blocking condition (new cue/same reward) and an "upwards unblocking" condition (new cue/bigger reward) to allow the authors to determine whether cells coded for value or for some other aspect of the stimuli. A second experiment was performed that missed off the initial cue such that essentially the authors could look for learning of value coding.

This is a nice follow up to the first published study. There is new information here to reinforce the authors' conclusions that value coding is not the predominant for observable in this particular population of lateral OFC cells in rodents. The discussion about why this does not fit with some others' views on OFC coding is excellent.

I only have a small number of comments to help clarity.

1) The key issue of the study is the proportion of value coding neurons. I wonder whether it would be useful to plot this explicitly in a pie chart or similar so the reader can quickly see this. Equally, would not a slightly less biased route be to ask: is there evidence for linear encoding (positive/negative) across the three reward sizes in any recorded neuron? I doubt this will change the conclusion – it might even strengthen it.

2) It took a long time for me to figure out what was happening in Figure 4. Something other than making the initial odor cloud grey should be put in to make it clear that it wasn't presented.

3) It would be better if Figure 4 was presented in a way that represented the trial types rather than "preferred" and "non-preferred".

4) This isn't really an issue for the study and the conclusions, but I was surprised the downwards unblocking condition is being sold as a value reduction when the surprising thing in Holland's work is that it seems to confer positive value to the new stimulus in spite of the reduction in reward size. Any comments? Could this be related to the fact that the initial stimulus in the compound is shared across all trial types meaning it loses any predictive strength?

*Reviewer #2:*

This is an interesting study of cue-outcome association and value coding in the rat anterior insula/lateral orbitofrontal cortex (lOFC). It follows up closely on a paper with an ingenious behavioural manipulation published by the same group in *eLife* in 2014 (McDannald et al.,) that argued that rat lOFC neuron activity is not easily explained as simply encoding value of a reward outcome but aspects of the association between a stimulus and a reward outcome. Here a further ingenious twist is added by the use of a combination of both upshift and downshift procedures whereby some animals learn that some cues indicate an increase or a decrease in previously experienced reward levels. Some lOFC neurons are preferentially active in response to upshift cues, some to downshift cues, and some to both. Once again the message is that it is difficult to reduce the activity of neurons in this region simple to value encoding.

I have the following minor observations. With the exception of point 1 they mostly concern the Discussion.

1) Presumably the population of neurons that are not the main focus of the current study but which responded to all cues do not contain a mixture of neurons with either positive or negative encoding of value but that such effects even out when the activity of the entire population is plotted (Figure 2—figure supplement 1)?

2) In general the Discussion is thoughtful. Unusually for a rodent neurophysiology study there is a serious attempt to engage with human neuroimaging and macaque neurophysiology data in a sophisticated way. The argument at the end of the Discussion is, however, difficult to follow and it is very salient because it is the final paragraph of the manuscript and thus seems like the authors' conclusion. It is difficult, for example, to square some of the manuscript's empirical findings with this paragraph. For example, it is difficult to square the prominence of downshift responses reported here in a simple and direct manner with neurons with activity that covaries with value but this seems to be authors' argument.

I agree that it is possible to imagine a brain that has both types of neurons and that they might operate in relation to one another. However, the authors seem to go further and suggest that the neurons they study here and the value neurons reported previously may be one and the same in nature. Given that the value effects that are reported in some previous macaque studies and human neuroimaging studies are not ubiquitous across the ventromedial and orbitofrontal cortex doesn't this already mean that are other types of neurons in other parts of the primate ventromedial and orbitofrontal cortex some of which may be more similar to the neurons the authors study here? There may be many disadvantages to neuroimaging but one advantage is that it is possible to get a sense of activity across the brain and when an attempt is made to look for both value and cue-outcome association representations in the same study (e.g. Noonan et al., J. Neuroscience, 2011) the results implicate different subdivisions of lOFC and ventromedial prefrontal cortex.

The section is also little unusual in the way that it attributes views to another author but cites a review written jointly by that person and one of the authors of the current report. There is something a little unusual about the style in which this section is written.

---

## [Author Response]

Reviewer #1:

*1) The key issue of the study is the proportion of value coding neurons. I wonder whether it would be useful to plot this explicitly in a pie chart or similar so the reader can quickly see this. Equally, would not a slightly less biased route be to ask: is there evidence for linear encoding (positive/negative) across the 3 reward sizes in any recorded neuron? I doubt this will change the conclusion – it might even strengthen it.*

We have addressed this comment in two ways:

We have added a pie chart to Figure 2 as Figure 2, followed by the reference to this figure in the text;

“Thus, if we disregard the very small number of neurons showing linear coding of value, our results here largely replicated what we observed when unblocking learning by shifting value versus identity (Figure 2).”

We also investigated positive/negative linear encoding across the three reward sizes in any neuron. For this analysis, we did not exclude cells with a baseline firing rate above 10 Hz, nor did we limit our analysis to odor-responsive cells. Repeating our screen across all cells, we found a total of 6 value-coding cells, one more than we had found previously.

“Few neurons exhibited firing that clearly rank ordered the cues according to their value (Figure 2; 5/60; 8%). […] suggesting that our screen had not prevented us from finding value-coding cells.”

*2) It took a long time for me to figure out what was happening in Figure 4. Something other than making the initial odor cloud grey should be put in to make it clear that it wasn't presented.*

Thank you for your thoughtful suggestion. We have modified Figure 4 and the corresponding caption to be clearer that no odor was presented.

*3) It would be better if Figure 4 was presented in a way that represented the trial types rather than "preferred" and "non-preferred".*

In Figure 4, we have changed the figure to present the trial types as we do in Figure 4: with other1 and other2, whose identities are indicated in the caption. Rather than a preferred response, we have indicated the tuned response.

*4) This isn't really an issue for the study and the conclusions, but I was surprised the downwards unblocking condition is being sold as a value reduction when the surprising thing in Holland's work is that it seems to confer positive value to the new stimulus in spite of the reduction in reward size. Any comments? Could this be related to the fact that the initial stimulus in the compound is shared across all trial types meaning it loses any predictive strength?*

While the above comment is true, our understanding (from the literature and discussions with Dr. Peter Holland) is that it applies primarily to decreases in reward number. In fact this is one reason why we moved from a number downshift to a size downshift in this experiment. This dichotomy is evident in published work. For example, in Esber and Holland (EJN, 2014), the rats experience concentration and number downshifts. While rats show excitation to the number downshift cue, they do not show excitation to the concentration downshift cue. Dr. Holland’s explanation as we understand it is that excitation may develop in response to a number downshift because the first reward is excitatory; thus, rats go to the well because they will receive the first expected reward. There is a delay before the subsequent reward is omitted, so this has a weaker associative effect on the number downshift cue since several seconds pass between the cue and a reward downshift. However, concentration is not excitatory. When rats first receive a concentration downshifted outcome, they immediately know which outcome they received and that it is worse than the previous outcome. This has a stronger associative effect because there is less delay between the cue and the downshifted outcome. Dr. Holland says that he has found similar effects for changes in size (personal communication); thus the excitatory effect seems to be unique to changes in number of rewards and perhaps only when presented sequentially.

In any event, we specifically chose a size change to avoid this issue. Fluid is dispensed in a fraction of a second, and rats immediately know that they received a smaller outcome. For this reason, we think that size in our design is more like concentration than number based on the immediacy of the outcome following the cue. Consistent with this goal, our rats did not show excitation to the downshifted cue; rather they showed inhibition of behavior – slower latencies. So we think in our particular case, that this concern does not apply.

However, because Peter and others have indeed found excitation in response to cues predictive of downshift, we will clarify this point in our Introduction. We have copied where we will elaborate on this difference between concentration and number below.

“Here we have designed a follow-up experiment to test this question. In this experiment, we have copied the design of the prior study (McDannald et al., 2014), recording OFC neurons during unblocking with two changes. […] This change will enable us to differentiate between the upshifted and downshifted cues behaviorally because the former will lead to excitation and the latter inhibition.”

Reviewer #2:

*1) Presumably the population of neurons that are not the main focus of the current study but which responded to all cues do not contain a mixture of neurons with either positive or negative encoding of value but that such effects even out when the activity of the entire population is plotted (Figure 2—figure supplement 1)?*

Thank you for raising this question. We screen for value in cells that respond to all cues and consider cells that respond to all four cues with value encoding to be value-signaling cells and not putative sensory cells as in Figure 2—figure supplement 1. However, we looked through the manuscript for the description of how we screened for these cells and found that this was buried in the methods: “Neurons were classified as putative sensory neurons and excluded from further consideration if they significantly increased firing to all odors and were not value-coding.”

Below is a revised caption. Thus, it will be clear in looking at this figure that value-coding cells are not included in this population:

“Figure 2—figure supplement 2. Odor-responsive units not analyzed in the main text […] d)52/174 (30%) of the excitatory odor-responsive cells showed a significant phasic response to each of the four odor cues. *Neurons were classified as putative sensory neurons only if they were not value-coding.* These neurons fired to all of the odors, even the blocked odor, thus their firing cannot be easily explained as signaling information about the predicted outcomes. However they might be signaling information about the cues themselves, such as their shared sensory features or intrinsic salience. When we examined the firing of neurons in this population on the first 10 trials of unblocking,”

Further, we noticed that our descriptions of how we screened for value cells in the main text did not indicate this either and updated the text accordingly:

“Our primary goal in the current experiment was to test whether upshift-responsive neurons, identified in our prior study, exhibit a firing pattern consistent with value signaling, or whether they might be signaling features. […] We isolated cells that responded significantly above baseline to the upshift odor but were not value-coding.”

*2) In general the Discussion is thoughtful. Unusually for a rodent neurophysiology study there is a serious attempt to engage with human neuroimaging and macaque neurophysiology data in a sophisticated way. The argument at the end of the Discussion is, however, difficult to follow and it is very salient because it is the final paragraph of the manuscript and thus seems like the authors' conclusion. It is difficult, for example, to square some of the manuscript's empirical findings with this paragraph. For example, it is difficult to square the prominence of downshift responses reported here in a simple and direct manner with neurons with activity that covaries with value but this seems to be authors' argument.*

I agree that it is possible to imagine a brain that has both types of neurons and that they might operate in relation to one another. However, the authors seem to go further and suggest that the neurons they study here and the value neurons reported previously may be one and the same in nature. Given that the value effects that are reported in some previous macaque studies and human neuroimaging studies are not ubiquitous across the ventromedial and orbitofrontal cortex doesn't this already mean that are other types of neurons in other parts of the primate ventromedial and orbitofrontal cortex some of which may be more similar to the neurons the authors study here? There may be many disadvantages to neuroimaging but one advantage is that it is possible to get a sense of activity across the brain and when an attempt is made to look for both value and cue-outcome association representations in the same study (e.g. Noonan et al., J. Neuroscience, 2011) the results implicate different subdivisions of lOFC and ventromedial prefrontal cortex. The section is also little unusual in the way that it attributes views to another author but cites a review written jointly by that person and one of the authors of the current report. There is something a little unusual about the style in which this section is written.

We apologize for the confusion. The intent of the final paragraph is indeed a bit muddled. We really only meant to point out that at least one of the proponents of the OFC=value hypothesis – Camillo Padoa-Schioppa – has described in writing a definition of value that seems synonymous (to us) with the type of value that would be a specific attribute of the “goods”. Thus while the data and a how they are described refer to value as independent of the specific features of the learned associations, the writings and the statements this person makes in public are in fact different. In the “dialogue” that we cite, Dr. Padoa-Schioppa in fact states explicitly that the value signaled by OFC neurons is an attribute of the good or outcome. And previously he has written that devaluation is the iconic economic decision… (we cite both pieces). So we simply wanted to close by pointing out that IF this is what is meant by “neuroeconomic value”, *then* it is not at odds with our definition or our findings. We have tried to recraft this final paragraph (see below) to better reflect this intention:

“Finally it is worth noting that neuroeconomic value may not be the same sort of value identified in our lexicon as pure, abstract, general or cached (i.e., independent of outcome features). […] As long as what is being signaled by such activity remains an attribute or feature of the predicted outcome – and sensitive to changes in that feature without further learning – then this would be consistent with our current data.”